# Models and Theoretical Analysis of SoOp Circular Polarization Bistatic Scattering for Random Rough Surface

**Xuerui Wu** [1,2,3,*] **and Shuanggen Jin** [2,3]

[1] School of Resources, Environment and Architectural Engineering, Laboratory of National Land Space Planning and Disaster Emergency Management of Inner Mongolia, Chifeng University, Chifeng 024000, China

[2] Shanghai Astronomical Observatory, Chinese Academy of Sciences, Shanghai 200030, China; sgjin@shao.ac.cn

[3] Shanghai Key Laboratory of Space Navigation and Positioning Techniques, Shanghai 200030, China

[*] Correspondence: xrwu@shao.ac.cn; Tel.: +86-21-34775291; Fax: +86-21-64384618

**Abstract:** Soil moisture is an important factor affecting the global climate and environment, which can be monitored by microwave remote sensing all day and under all weather conditions. However, existing monostatic radars and microwave radiometers have their own limitations in monitoring soil moisture with shallower depths. The emerging remote sensing of signal of opportunity (SoOp) provides a new method for soil moisture monitoring, but only an experimental perspective was proposed at present, and its mechanism is not clear. In this paper, based on the traditional surface scattering models, we employed the polarization synthesis method, the coordinate transformation, and the Mueller matrix, to develop bistatic radar circular polarization models that are suitable for SoOP remote sensing. Using these models as a tool, the bistatic scattering versus the observation frequency, soil moisture, scattering zenith angle, and scattering azimuth at five different circular polarizations (LR, HR, VR, + 45° R, and −45° R) are simulated and analyzed. The results show that the developed models can determine the optimal observation combination of polarizations and observation angle. The systematic analysis of the scattering characteristics of random rough surfaces provides an important guiding significance for the design of space-borne payloads, the analysis of experimental data, and the development of backward inversion algorithms for more effective SoOP remote sensing.

**Keywords:** SoOP; circular polarization; bistatic scattering; remote sensing

## 1. Introduction

Soil moisture is an important factor affecting the global climate and environment. Its existence in the earth system and its spatial transmission mode play a vital role in the global energy balance. It controls the exchange of hydrothermal energy between the land and the atmosphere. Related studies have shown that there is a strong feedback relationship between soil moisture and abnormal climate. Soil moisture is an important index parameter in the fields of hydrology, meteorology, and agricultural scientific research. Large-scale soil moisture monitoring and retrieval is an important part of agricultural research and ecological environment assessment. At the same time, soil moisture is also the link between surface water and groundwater, and it is an important part of the land surface ecosystem and water cycle. Therefore, soil moisture information plays an important role in improving regional and even global climate model forecasts, global water cycle laws, water resource management,

watershed hydrological models, crop growth monitoring, crop yield estimation, environmental disaster monitoring, and other related natural and ecological environmental issues [1].

Traditional observation methods mainly use observations at discrete stations or corresponding meteorological stations, which can only represent a limited observation area (≈10–100 cm). They are also time-consuming and labor-intensive and cannot meet the needs of large-scale and high-efficiency soil moisture observation. Using this traditional monitoring method makes it difficult to match the corresponding weather and hydrological model (0.1–10 km) in terms of spatial scale and time accuracy, so this method cannot effectively study the effect of soil moisture on environmental changes.

Remote sensing methods can obtain soil moisture information with high efficiency and at a large scale. Optical, infrared, and microwave remote sensing are the main remote sensing methods for earth observation. The corresponding sensors work in the visible, infrared, and microwave bands of the electromagnetic spectrum, but these remote sensing methods have their own limitations. Optical and infrared remote sensing are limited to weather conditions and cannot work all day and under all weather conditions; microwave remote sensing overcomes this shortcoming and has the advantages of all-day, all-weather, and strong penetration. The real part of the dielectric constant of water is 80, and that of dry soil is 3.5. An increase in soil moisture will cause an increase in the dielectric constant, and thus a decrease in emissivity or an increase in reflectivity. The basic principle of microwave remote sensing soil moisture detection is the large dielectric constant difference between water and dry soil [2]. Among them, the P-band and L-band are particularly favorable for soil moisture observation. In these bands, the atmospheric attenuation decreases, and the vegetation penetration increases.

Traditional active and passive microwave methods (microwave radiometer and radar) have their own advantages and disadvantages. The radiometer measures the surface brightness temperature, and then can use the emissivity information to retrieve soil moisture. Radiation measurement is not sensitive to surface roughness, but is easily affected by background brightness and artificial RFI (Radio Frequency Interference). Its spatial resolution is high, while data processing is simple, but its time resolution is low. Compared with the radiometer, in the radar, the larger the soil moisture content, the larger the backscattering coefficient, the lower the sensitivity of the single station radar to the soil moisture, and the more easily the backscattering is affected by surface characteristics, such as surface roughness, soil dielectric constant, and vegetation structure. Its data processing is complex and the spatial resolution is low [3]. The unique observation geometry model of the bistatic radar has become a new method and technology for remote sensing monitoring of soil moisture and vegetation. However, in the general sense, bistatic radars need to develop special transmitters and receivers, which have limitations such as high cost, heavy load, and low power consumption.

The emerging signal of opportunity (SoOP) technology uses the existing navigation satellite group or communication satellites as the signal transmission sources, and only needs to develop a special reflected signal receiver to achieve effective monitoring of soil moisture in bistatic radar mode [4–6]. Opportunity signal reflection remote sensing provides new opportunities for root zone soil moisture acquisition in the P-band. The P-band penetration has an advantage, with a penetration depth of about 40 cm at soil moisture of no more than 2–3 volumetric % when there is no vegetation cover, while its penetration depth is 10–15 cm and L-band signal is no more than 5 cm with vegetation presence [7]. Because digital communication satellites are used as signal sources, there is no need to develop a special transmitter. Therefore, P-band opportunity signal reflection remote sensing has many advantages, such as low cost, low power consumption, cheap price, and high spatiotemporal resolution [8].

GNSS-R (Global Navigation Satellite System-Reflectometry) uses navigation satellites as its signal source and uses its reflected signals to remotely sense ground feature parameters [9]. At present, the earliest and more extensive study of GNSS-R technology on land surface is remote sensing of soil moisture, and the existing research is mostly carried out from an experimental perspective [10–12].

Most of the existing scattering models were aimed at the backscattering of a monostatic radar, or the radiation characteristics of passive microwaves. Studies focusing solely on SoOP scattering characteristics have been paid less attention, and most of them are concentrated in one

plane [13,14]. Although recent studies have pointed out that scattering azimuth will affect the polarization characteristics of bare soil, only linear polarization characteristics were considered in the model, and relatively few studies on SoOP circular polarization characteristics have been made. In order to overcome the influence of the ionosphere, the signals were transmitted by navigation satellites' RHCP (Right Hand Circular Polarization) signal. After the signal is reflected from the ground, the polarization characteristics will change. However, making full use of its polarization characteristics is an urgent problem for SoOP application. At the same time, research on the ocean also found that the theoretical simulation of the co-polarized scattering component and the actual waveform were poorly matched, and a theoretical model needs to be established to simulate and analyze the co-polarized scattering component. Relevant research on the parameters of bare soil, especially the bistatic radar scattering mechanism model that focuses on various polarizations, including circular polarization characteristics, has been studied less. Due to the lack of the mechanism model, the full polarization of its bistatic scattering (circular and linear polarization) lacks awareness of sensitivity, which limits the further development of the technology. Therefore, for SoOP soil moisture remote sensing, it is important to fully excavate the polarization characteristic information of navigation satellite signals. Carrying out the circular polarization theory research of the bistatic radar scattering model is important for space-borne load design, experimental data analysis, and backward inversion algorithm development [15].

In this paper, we will develop the surface scattering model to simulate the bistatic scattering characteristics at circular polarization for SoOP applications. In Section 2, the theoretical formulations are presented. The simulations and analysis are given in Section 3. In Section 4, discussion is shown, and finally the conclusions are presented in Section 5.

## 2. Models and Theory

### 2.1. Models Description

This paper will use the random rough surface scattering models to establish the mathematical relationship between the electromagnetic parameters of the opportunity signal and the physical and geometrical parameters of the surface. The electromagnetic parameter of the opportunity signal system is the bistatic radar cross section. For bare ground, the physical geometric parameters refer to the dielectric constant and surface roughness of the soil. Soil is a dielectric mixture of air, solid soil, bound water, and free water [16,17]. Each component has an important influence on the soil dielectric constant. The roughness characteristics of a random surface can be expressed by the root mean square height and the correlation length. These two parameters define the surface roughness from vertical and horizontal scales, respectively. For SoOP remote sensing, the soil dielectric constants of the bare surface and the surface roughness are coupled to each other, and it is generally difficult to distinguish which change caused the change in the sensor.

Commonly used stochastic surface scattering theoretical models include the KA (Kirchhoff Approach) model, SPM (Small Perturbation Method) model, IEM (Integrated Equation Model) model, and AIEM (Advanced Integrated Equation Model) model, which was further improved [18,19]. Loosely speaking, the GO (Geometrical Optics) model is best suited for very rough surfaces, the PO (Physical Optics) model is suitable for intermediate roughness surfaces, and the SPM model is suitable for surfaces with short correlation lengths [20].

In fact, the roughness of the natural surface is continuous, including various levels of roughness levels. To reproduce the bistatic scattering characteristics of different rough surfaces, continuous models are required to scatter the natural surface conditions under different roughness conditions. With characteristic simulation, the AIEM model can more closely approximate the action of electromagnetic waves on actual surface conditions.

### 2.2. Coordinate System Transformation

In the process of improving the model, it is also necessary to perform coordinate conversion from the original BSA (Backward Scatter Alignment) coordinate system to the FSA (Forward Scatter Alignment) coordinate system. That is, to the FSA coordinate system shown in Figure 1 [20].

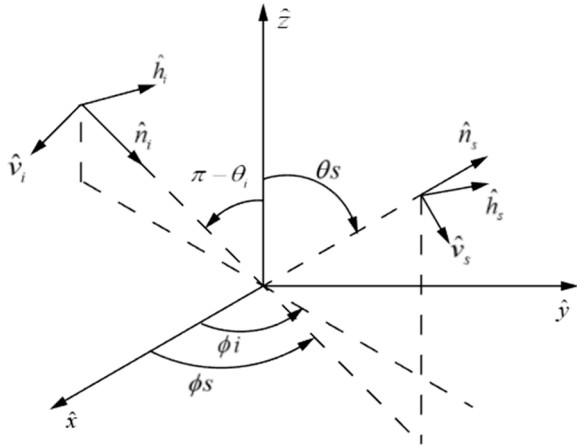

**Figure 1.** Geometry for the Forward Scatter Alignment (FSA) coordinate system.

The unit vectors in the figure are defined as follows.

$$\hat{n}_i = \hat{x} \sin \theta_i \cos \phi_i + \hat{y} \sin \theta_i sin\phi_i + \hat{z} cos\theta_i \tag{1}$$

$$\hat{h}_i = -\hat{x} sin\phi_i + y cos\phi_i \tag{2}$$

$$\hat{v}_i = \hat{h}_i \times \hat{n}_i \tag{3}$$

$$\hat{n}_s = \hat{x} \sin \theta_s \cos \phi_s + y \sin \theta_s \sin \phi_s + z \cos \theta_s \tag{4}$$

$$\hat{h}_s = \hat{x} \sin \phi_s + \hat{y} \cos \phi_s \tag{5}$$

$$\hat{v}_s = \hat{h}_s \times \hat{n}_s \tag{6}$$

The only difference between the BSA and FSA is that $\hat{h}_s$ is replaced with $-\hat{h}_s$.

### 2.3. Polarization Synthesis

For the GO, PO, and SPM models, the existing form is a backscattering model for a monostatic radar. The AIEM model has a bistatic radar scattering model for linear polarization. Therefore, in order to develop a bistatic radar circular polarization scattering model suitable for SoOP remote sensing, this study improves the original model. Generally speaking, we use the polarization synthesis formula shown below [21].

$$\sigma_{rt}^o(\psi_r, \chi_r, \psi_t, \chi_t) = 4\pi \widetilde{Y^r} I_p M Y^t \tag{7}$$

With this formula (Equation (7)), the bistatic scattering cross section for any combinations of transmitted and received polarizations can be calculated, where the subscripts $t$ and $r$ are the transmitted and received polarizations, respectively; $Y^t$ and $Y^r$ are the normalized Stokes vectors characterizing the transmitter and receiver polarizations, respectively.

$$Y^t = \begin{bmatrix} 1 \\ \cos 2\psi_t \cos 2\chi_t \\ \sin 2\psi_t \cos 2\chi_t \\ \sin 2\chi_t \end{bmatrix} \text{ and } Y^r = \begin{bmatrix} 1 \\ \cos 2\psi_r \cos 2\chi_r \\ \sin 2\psi_r \cos 2\chi_r \\ \sin 2\chi_r \end{bmatrix}, \tag{8}$$

where $(\psi_t, \chi_t)$ and $(\psi_r, \chi_r)$ are the orientation and ellipticity angles for the transmitted and received polarizations. $I_p$ is the diagonal matrix required for coordinate transformation.

$$I_p = \begin{bmatrix} 1 & 0 & 0 & 0 \\ 0 & 1 & 0 & 0 \\ 0 & 0 & -1 & 0 \\ 0 & 0 & 0 & -1 \end{bmatrix} \tag{9}$$

The matrix M is the Mueller matrix and it defines as the following:

$$M = \widetilde{R}^{-1} W R^{-1} \tag{10}$$

$$W = \begin{bmatrix} S_{vv}S_{vv}^* & S_{vh}S_{vh}^* & S_{vv}S_{vh}^* & S_{vh}S_{vv}^* \\ S_{hv}S_{hv}^* & S_{hh}S_{hh}^* & S_{hv}S_{hh}^* & S_{hh}S_{hv}^* \\ S_{vv}S_{hv}^* & S_{vh}S_{hh}^* & S_{vv}S_{hh}^* & S_{vh}S_{hv}^* \\ S_{hv}S_{vv}^* & S_{hh}S_{vh}^* & S_{hv}S_{vh}^* & S_{hh}S_{vv}^* \end{bmatrix} \tag{11}$$

$$R = \begin{bmatrix} 1 & 1 & 0 & 0 \\ 1 & -1 & 0 & 0 \\ 0 & 0 & 1 & 1 \\ 0 & 0 & -j & j \end{bmatrix} \tag{12}$$

where $\widetilde{R}$ means R transpose; $S_{rt}$ is the specific component of the scattering matrix, S; while the subscripts, t and r, stand for the polarizations of the transmitted and received signals, respectively.

*2.4. Model Validation*

The model developed in this paper is based on the original backward monostatic radar model, and the developed bistatic scattering models can calculate various polarizations using the method of polarization synthesis. For the verification of the developed model, the method of the 'model verification model' is adopted. The model to be developed is set to the direction of backscatter, and the original model is compared to verify the correctness of the bistatic scattering models. For different polarizations, the orientation and ellipticity angles of the modified Stokes vector were modified to linear angles and compared with the model without polarization synthesis to verify the circular polarization scattering characteristics. The modified Stokes vector forms for different polarizations are shown in the Table 1 [21].

**Table 1.** Modified Stokes vectors for different polarizations.

|  | V pol. | Hpol | LHCP pol | RHCP pol | +45° pol | −45° pol |
|---|---|---|---|---|---|---|
| Modified Stokes Vectors | $\begin{bmatrix} 1 \\ 0 \\ 0 \\ 0 \end{bmatrix}$ | $\begin{bmatrix} 0 \\ 1 \\ 0 \\ 0 \end{bmatrix}$ | $\begin{bmatrix} 0.5 \\ 0.5 \\ 0 \\ 1 \end{bmatrix}$ | $\begin{bmatrix} 0.5 \\ 0.5 \\ 0 \\ -1 \end{bmatrix}$ | $\begin{bmatrix} 0.5 \\ 0.5 \\ -1 \\ 0 \end{bmatrix}$ | $\begin{bmatrix} 0.5 \\ 0.5 \\ 1 \\ 0 \end{bmatrix}$ |

## 3. Simulation and Analysis

Using the model established by the developments above, we simulate the bistatic circular polarization response characteristics with different parameters. Here, we employ the models presented in papers [16,17] to calculate the dielectric constants of different soil moistures. The random rough surface scattering model provides a mechanism for the sensitivity analysis of surface parameters in SoOP remote sensing.

### 3.1. Frequency Response

When digital communication satellites are used as signal sources for opportunistic signal reflection remote sensing, they mainly work in the P-band. The navigation satellite system basically works in the low-frequency L-band to the high-frequency L-band. The Indian regional navigation satellite system implemented by the Indian Space Research Organization has three operating frequency bands: C-band, S-band, and L-band. As for the carrier frequency bands of the digital communication satellites and navigation satellite systems, the P-band is suitable for monitoring root zone soil moisture, and the L-band is suitable for monitoring near surface soil moisture. The frequency response of the random rough surface in different carrier bands is shown in Figure 2. In this figure, we have employed the models present in papers [16,17] to calculate the dielectric constants. The volumetric soil moisture content is 35%, the sand content is 10%, and the clay content is 60%. The relationship between the bistatic radar scattering cross section (BRCS) with frequency was simulated at 0.3–7 Ghz, the root mean square height was 0.45 cm, and the correlation length was 18.75 cm. The polarization of the transmitted signals were RHCP. The polarization characteristics of reflected signals are as follows: LHCP, H polarization, V polarization, +45° polarization, and −45° polarization, i.e., LR, HR, VR, +45°, and −45° R. It can be seen from the simulation that the changes of the five polarizations at different frequencies are not much different. As the frequency increases, BRCS first increased and then decreased, and there was a scattering peak at about 1.3 GHz. In the following simulations, we will use the P-band (0.3 Ghz) as an example to simulate the bistatic circular polarization scattering characteristics of the five polarizations with different surface geometric and physical parameters.

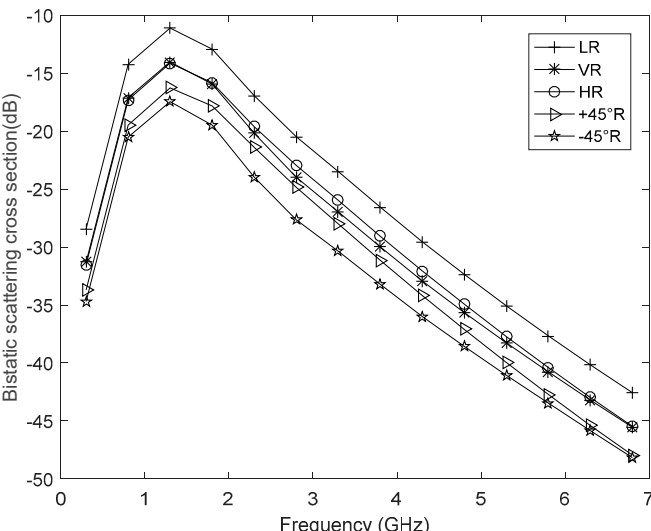

**Figure 2.** Variation of bistatic radar scattering cross section (BRCS) with frequency for five different polarizations.

### 3.2. Soil Moisture Response

The soil texture, roughness, soil temperature, and soil moisture will affect the dielectric constant change, and then the BRCS. By using the developed models, the response of different soil parameters to BRCS can be simulated and analyzed. Figure 3 simulates the change of BRCS with soil moisture at P-band, 30° incident angle, 5° reflection zenith angle, and 120° scattering azimuth angle. It can be seen from Figure 3 that as the soil moisture increases, the BRCS of the five polarizations increase. For the LR polarization in Figure 3a, the results simulated by the PO model and the AIEM model are the same, and the results simulated by the SPM model are basically the same as those simulated by the PO and AIEM models. However, under each soil moisture condition, the BRCS difference is about 10 dB. In Figure 3, the VR and HR polarizations have little difference in the simulation results using the three models, and the trends are basically the same. As for +45° R polarization, the results of the

SPM simulation and the PO and AIEM models have the largest difference. For −45° R, the difference between the simulation results of the three models is about 5 dB.

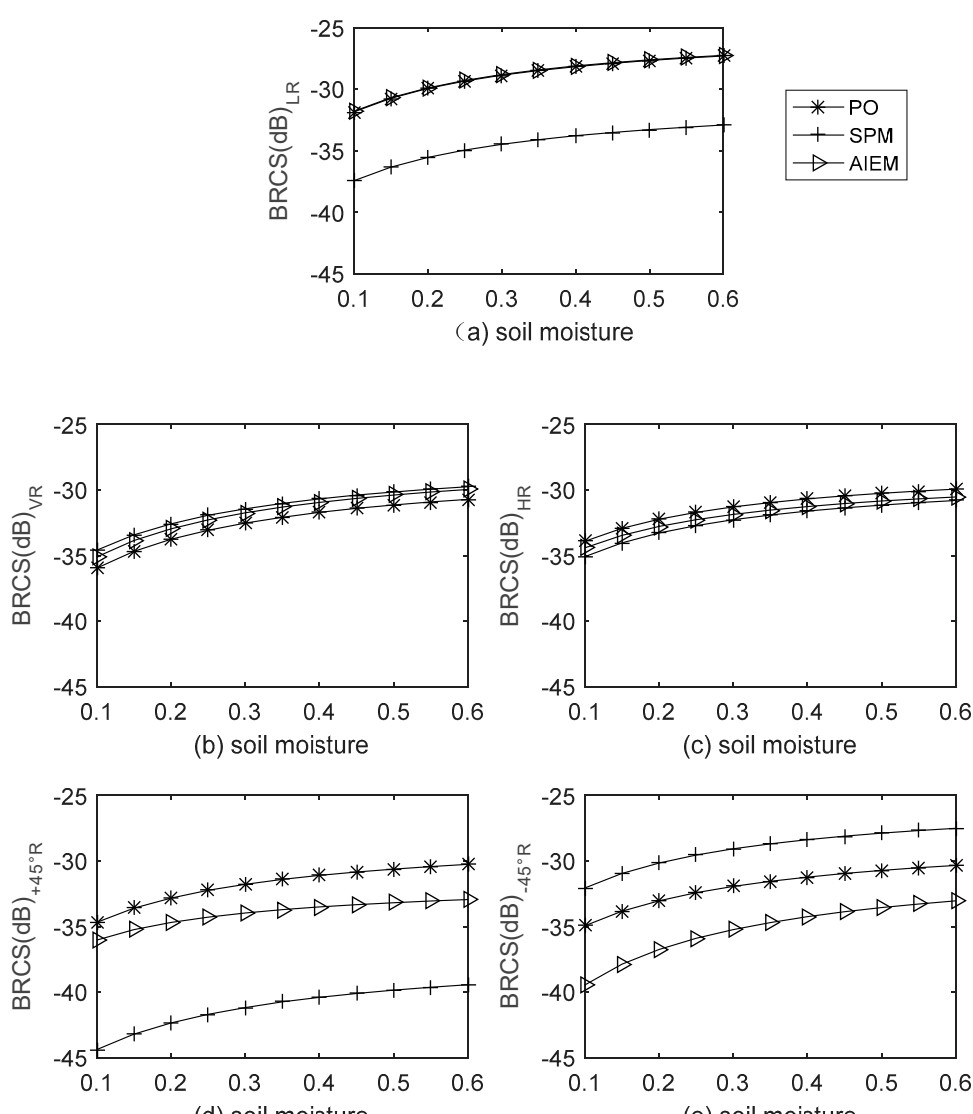

**Figure 3.** Soil moisture effects on BRCS at five different polarizations. $\theta_i = 30, \theta_s = 5, \varphi_s = 120$. Subfigures a to e represent the polarizations of LR, VR, HR, +45° R and −45° R, respectively.

We have simulated some other situations, including the specular plane, off-specular plane, and perpendicular plane. For these different scattering geometries, we compared the scattering properties with the soil moisture variations. From our simulations, we can see that as the soil moisture increases, the scattering values increase for the five different polarizations, no matter how the observation geometry changes. Here, we use Figure 4 as an illustration.

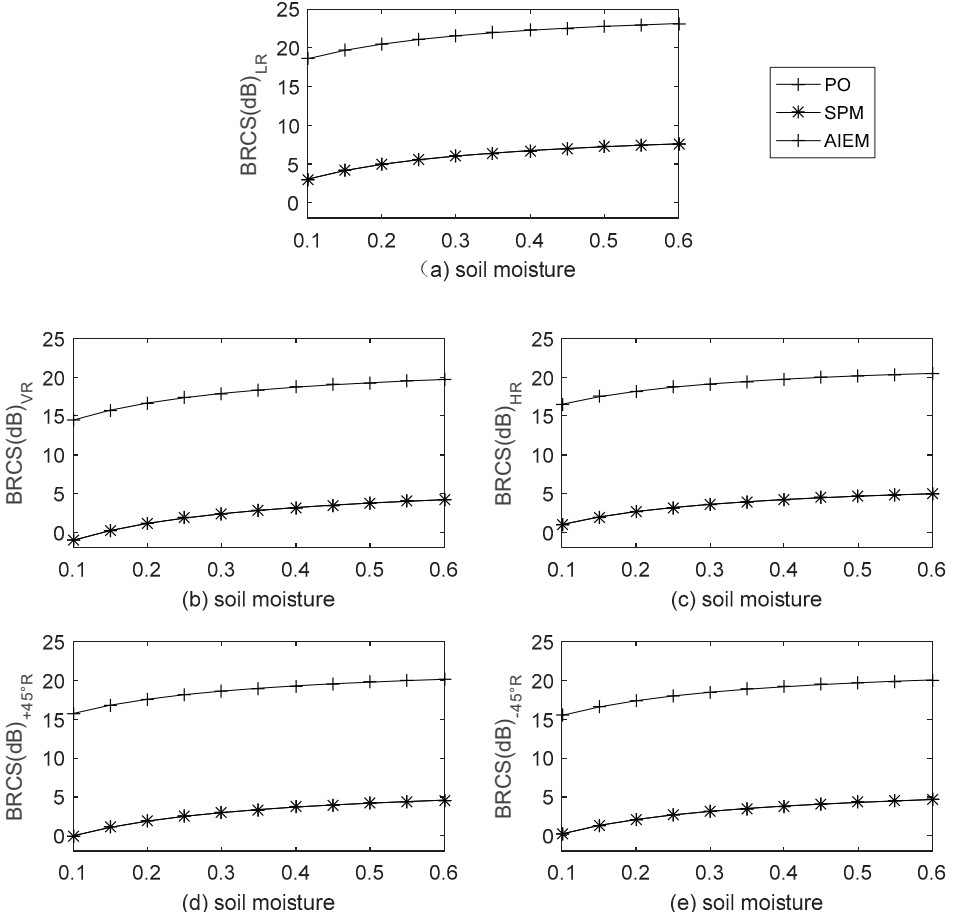

**Figure 4.** Soil moisture effects on BRCS at five different polarizations. $\theta_i = 30, \theta_s = 30, \varphi_s = 0$. Subfigures a to e represent the polarizations of LR, VR, HR, +45° R and −45° R, respectively.

From the simulations, as illustrated in Figure 4, we can see that the scattering values are the largest for the specular plane, which is also consistent with the maximum scattering value when the observation geometry is at the specular plane. However, no matter how the soil moisture changes, the scattering properties of BRCS increase with the soil moisture content for the five different polarizations. This phenomenon conforms to the existing theoretical law that more soil moisture results in larger dielectric constants and therefore leads to larger BRCS.

*3.3. Effect of Scattering Zenith Angle*

As can be seen in Section 3.2, we have illustrated the effects of soil moisture on the scattering properties. In this section, we use the models presented in paper [16,17] to get the dielectric constants. In order to focus on the effects of scattering geometry on scattering properties, we set the volumetric soil moisture content to a constant of 0.35, and the corresponding dielectric constant is about 19.44 + 4.91i.

Figure 5 shows the variation of BRCS with the scattering zenith angle at a 30° incident angle and 120° scattering azimuth angles. With the exception of +45° R, the rest of the polarizations are simulated using the three models: PO model, SPM model, and AIEM model. The results are similar. As the scattering zenith angle increases, BRCS decreases. When the scattering zenith angle is greater than 45°, the results of the PO model simulation are too small to be ignored, and the results of the AIEM simulation increase slightly after a large angle. As the scattering zenith angle is larger than 65°, the BRCS simulated by SPM model becomes very small. It can be seen from the simulation results that the effect of the AIEM model is better.

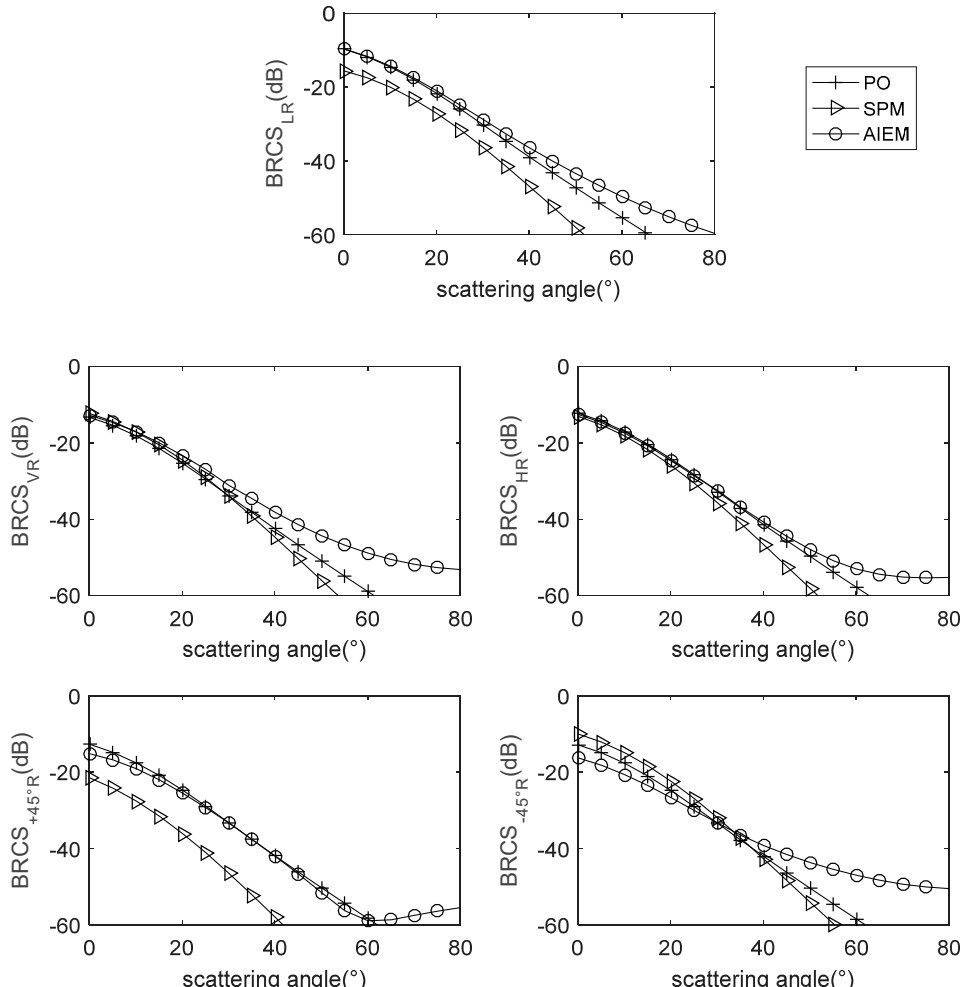

**Figure 5.** Scattering zenith angles' effects on BRCS at five different polarizations. $\theta_i = 30°$, $\theta_s = 30°$, $\varphi_s = 0°$. Subfigures a to e represent the polarizations of LR, VR, HR, +45° R and −45° R, respectively.

Figure 6 shows the variation of various polarizations with the scattering zenith angle when the scattering azimuth angle is 0°. As the incident energy and the scattered energy are in the same plane, a scattering peak appears when mirroring. This phenomenon can be clearly seen from Figure 6. With various polarizations, the BRCS increases first and then decreases with the scattering zenith angle. When the scattering zenith angle is about 30°, the BRCS value is the largest. Among the simulation results using the three models, the simulation results of the PO model and the SPM model are basically equal, and the AIEM model is slightly higher than the simulation results of the previous two models at each scattering zenith angle.

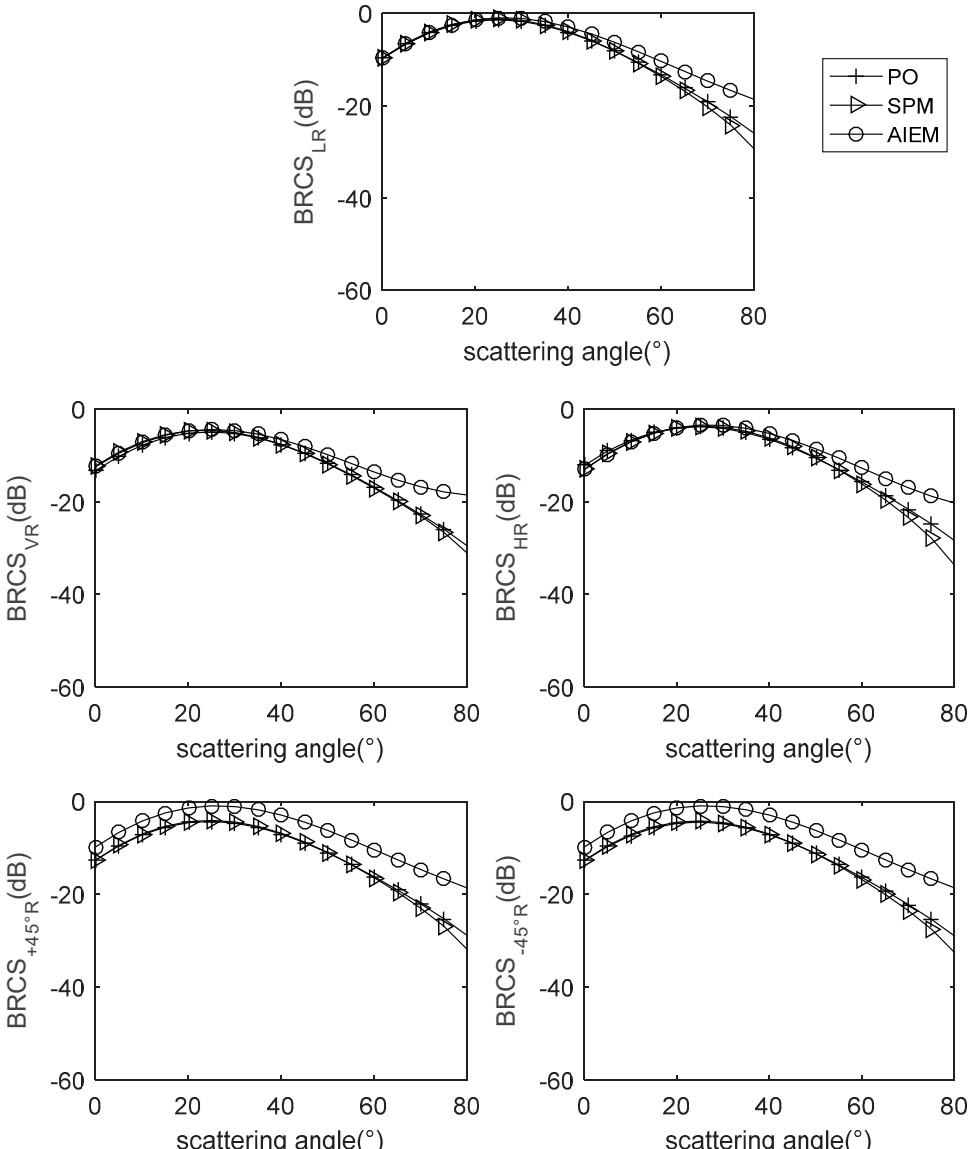

**Figure 6.** Scattering zenith angles' effects on BRCS at five different polarizations. $\theta_i = 30°$, $\varphi_s = 0°$. Subfigures a to e represent the polarizations of LR, VR, HR, +45° R and −45° R, respectively.

### 3.4. Effect of Scattering Azimuth Angles

Figure 7 shows the relationship between BRCS and the scattering azimuth, the incident angle is 20°, and the scattering angle is 40°. It can be seen from the figure that the BRCS changes with different polarizations are quite different. For LR, +45° R and −45° R, the scattering azimuth decreases and then increases with the increase of the scattering azimuth angles. The scattering values of the three polarizations have grooves at different scattering azimuth angles. RV and RH polarizations show single change trends with the increase of scattering azimuth angles.

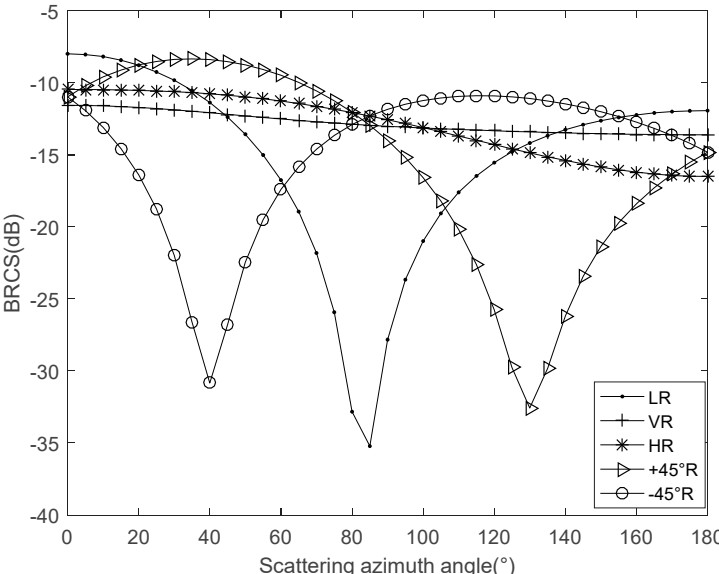

**Figure 7.** Scattering azimuth angles' effects on BRCS at five different polarizations.

### 3.5. Effects of Scattering Geometry and Soil Moisture

In this section, we will show the effects of different scattering geometries (scattering zenith angle and azimuth angels) and soil moisture contents on the scattering properties. We set the incidence angle 30°, while the scattering zenith angles vary from 0° to 85°, and the range of scattering azimuth angles is from 0° to 360°. In order to take the soil moisture variations into account, we illustrate three different soil moisture contents: volumetric soil moisture (vsm) of 0.1, 0.3, and 0.6. The roughness factors for these simulations are set constants, the rms height is 0.45 cm, and the correlation length is 18.75 cm. For the five different polarizations, the corresponding simulations are presented in Figures 8–12. Subfigures a–c in each figure are the BRCS for vms (0.1, 0.3, and 0.6). As for the subfigures d–f in each figure, we compare the BRCS differences for the different soil moistures. From the simulations, we can see that the scattering geometry (both scattering zenith angles and azimuth angles) will affect the final scattering properties to different extents. For the five different polarizations, the scattering properties are very different. There are the scattering peak values for the specular plane, but outside of this plane, the scattering properties vary at different extents. For the five different polarizations, the scattering properties are obviously different.

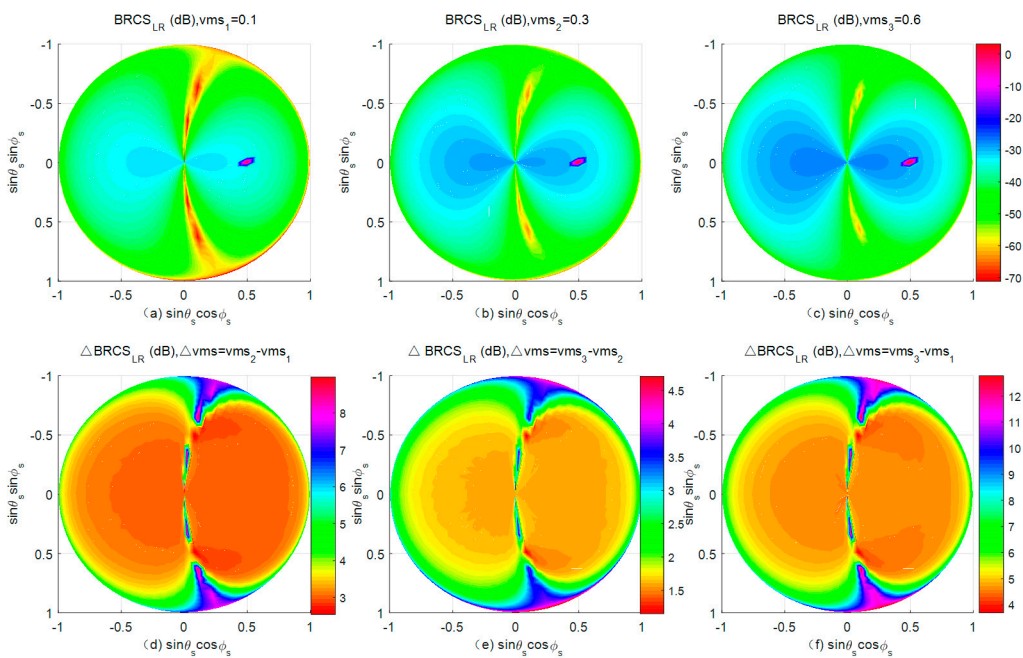

**Figure 8.** BRCS for different soil moisture contents (**a**–**c**) and the BRCS differences for different soil moisture differences (**d**–**f**) at LR polarization.

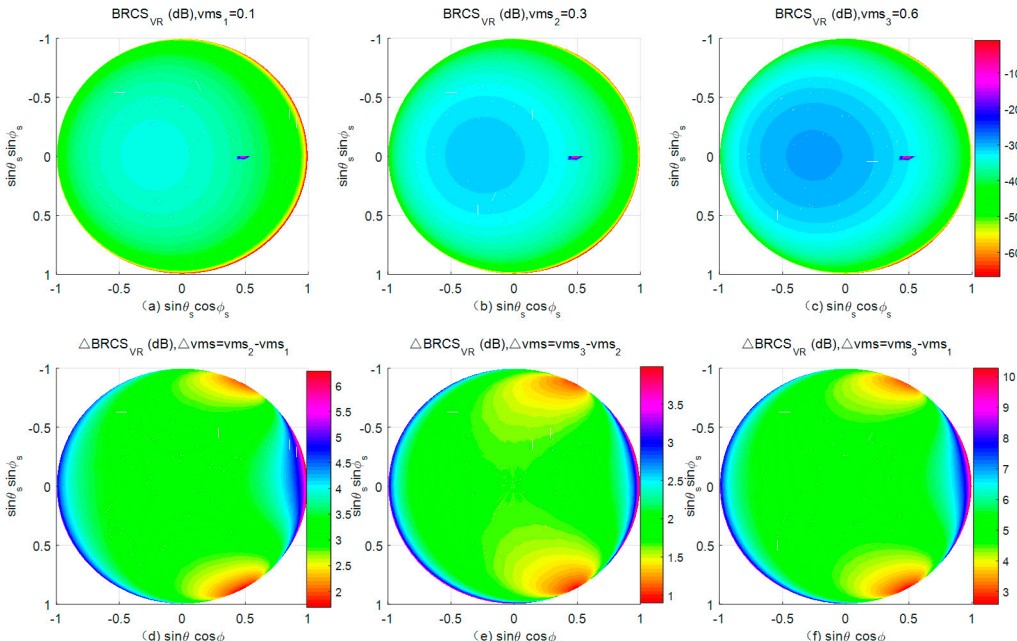

**Figure 9.** BRCS for different soil moisture contents (**a**–**c**) and the BRCS differences for different soil moisture differences (**d**–**f**) at VR polarization.

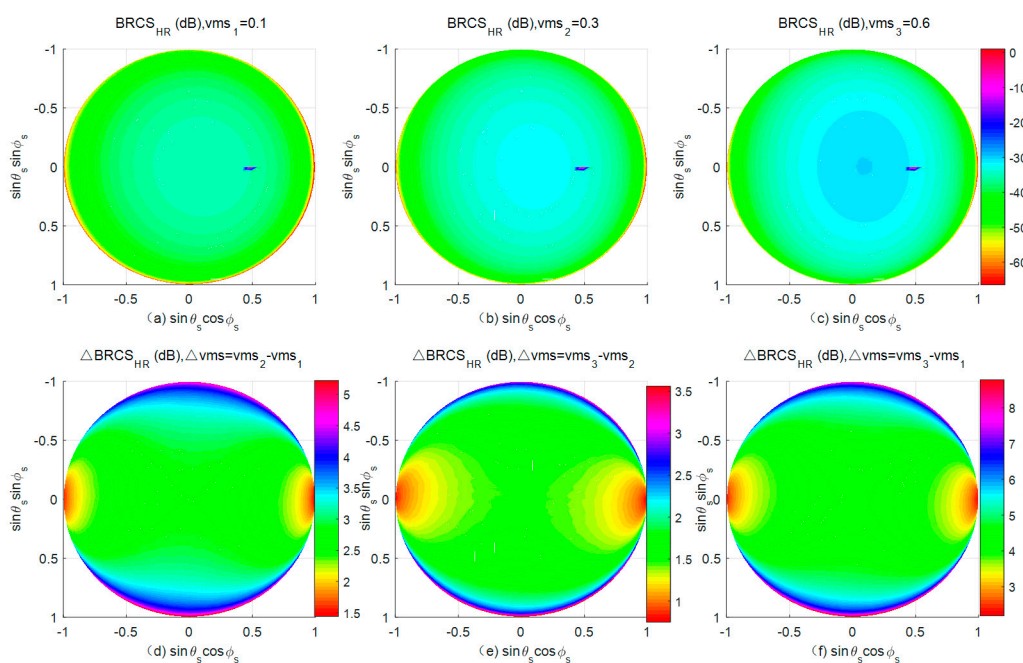

**Figure 10.** BRCS for different soil moisture contents (**a**–**c**) and the BRCS differences for different soil moisture differences (**d**–**f**) at HR polarization.

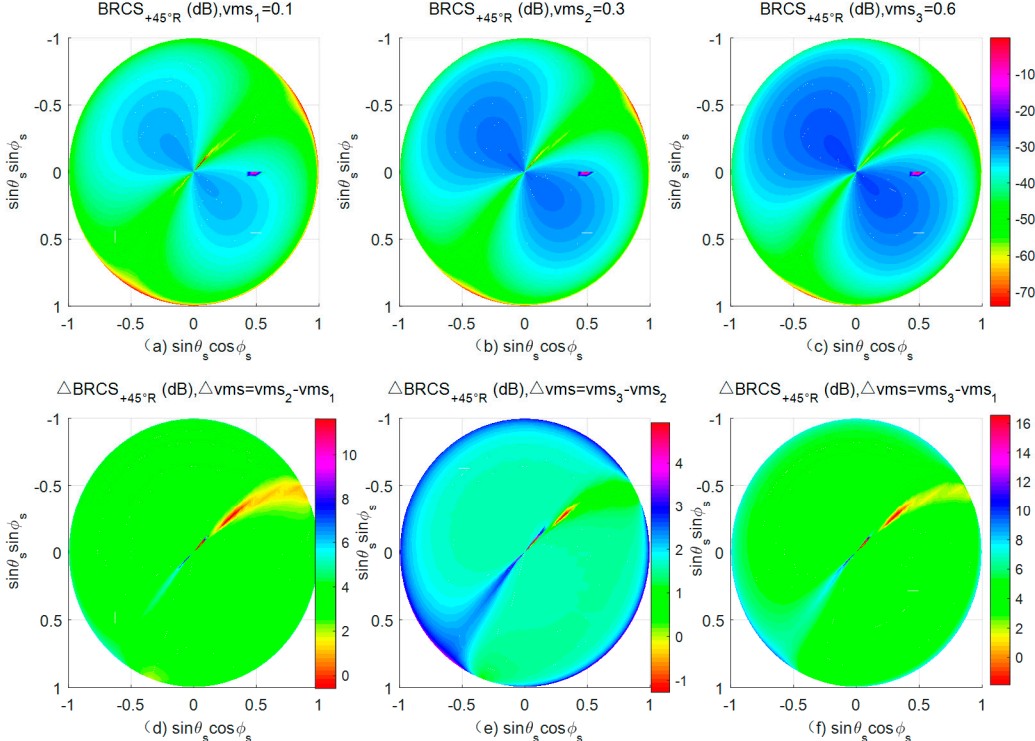

**Figure 11.** BRCS for different soil moisture contents (**a**–**c**) and the BRCS differences for different soil moisture differences (**d**–**f**) at +45° R polarization.

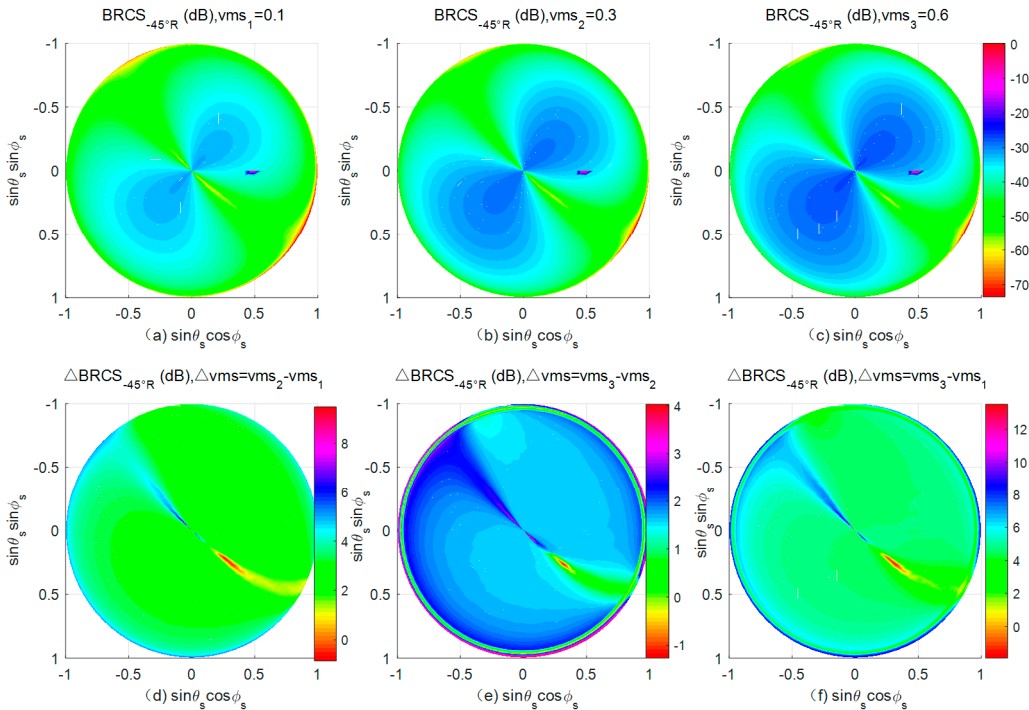

**Figure 12.** BRCS for different soil moisture contents (**a**–**c**) and the BRCS differences for different soil moisture differences (**d**–**f**) at −45° R polarization.

## 4. Discussion

As for GNSS-R remote sensing, it only takes the coherent scattering part at the specular direction for analysis. From the present analysis of the space-borne data, such as TDS-1 and CYGNSS, most are focused on the analysis of the coherent scattering from the first Fresnel zone, and always ignored the diffuse scattering powers. As for SoOP applications, the transmitter and the corresponding receivers form the typical bistatic-radar working mode, so the influence of the observation geometry on the scattering characteristics is critical. We can also see from our simulations that strong scattering values at the specular direction do exist, especially at the specular plane. This energy should be taken into account for future data analysis, but we should note that the random rough surface will not be very flat, and the surface roughness must result in the diffuse scattering. It can be seen from the analysis that the scattering value will peak at the mirror angle, but in this case the angle needs to be in a plane. When the scattering azimuth changes, the peak of the scattering disappears. It can also be seen through simulation, that the scattering value will have troughs at different scattering azimuths, but this scattering groove is related to the scattering azimuth of different polarizations. The scattering peaks and scattering grooves in the simulation analysis are of great significance for soil moisture inversion and are the angles that need attention in the subsequent soil moisture inversion. With the development of the algorithm and the improvement of the receiver's ability to receive signals, the bistatic scattering properties out of the specular plane must be taken good care of in the future analysis. Only by fully considering and effectively utilizing the bistatic scattering characteristics of various observation geometries, can we make better use of the advantages of SoOP remote sensing bistatic radar to observe the geophysical parameters. Our analysis and simulations give us the illustrations of these properties to some extent.

From our analysis, we can also see that the scattering properties vary a lot for different polarizations. The polarization mode is an important parameter for characterization of electromagnetic waves and is determined by the receiver antenna. Polarization is an important characteristic of electromagnetic waves. The polarization information of the surface reflection signal carries important information on the surface. The polarization ratio is an important information parameter for soil moisture inversion

and vegetation state research. Common polarization methods are linear polarization and circular polarization. The antenna polarization of SoOP remote sensing receivers are different, which causes the amplitude and phase characteristics of the target echo to be different, which will affect the detection sensitivity of the receiver. The model developed in this paper can simulate the reflection signal of arbitrary polarization. According to the simulation results, it was seen that the scattering characteristics of random rough surfaces were significantly different under different polarizations. Studying the target scattering characteristics has important guiding significance for SoOP antenna design. The calculation ability of our models for any polarizations will give guidance for future data analysis.

## 5. Conclusions

In the past two decades, we witnessed the promising development of GNSS-R remote sensing. With the same fundamentals, the SoOP technique employs the signals of a communication satellite system as the free transmitters, which has shown in recent years that SoOP is a promising remote sensing technique for the detection of geophysical parameters. With unique frequency advantages at the P-band, we employed SoOP for our analysis. The transmitters and the corresponding receivers of SoOP form the typically bistatic radar. Here, we simulated and analyzed the bistatic scattering properties at different scattering geometries (both scattering zenith angle and azimuth angles). Scattering peak values exist at the specular plane, and the information coming from the first Fresnel Zone should be taken, especially used as in the GNSS-R technique. Scattering grooves also exist, and this information could be useful for the development of soil moisture retrieval and vegetation corrections. From the simulations, we can see that these grooves were related to the scattering azimuth at different polarizations, which is related to other features of the development models. These models can simulate and analyze five different circularly polarized scattering characteristics of random rough surfaces, namely the transmitted polarization (RHCP) and the receiving polarizations (LHCP, H, V, +45°, and −45°) characteristics, respectively. Through the simulation analysis of the models, it was seen that different frequencies and soil moisture parameters can affect the circular polarization bistatic scattering characteristics. The scattering characteristics of the scattering zenith angle and scattering azimuth angle in the observation geometry at the five different polarizations were also different. We can find and determine the optimal observation combination by using different polarizations and different observation angles. It was also conducive to a more systematic and in-depth analysis of the scattering characteristics of random rough surfaces from a physical perspective, which will contribute to the design of more effective SoOP remote sensing inversion models in the future.

**Author Contributions:** X.W.—Conceptualization and original draft; S.J.—suggestions, review and revising. All authors have read and agreed to the published version of the manuscript.

**Funding:** This research was funded by National Natural Science Foundation of China (No. 41501384) and the Doctoral Scientific Fund Project of Chifeng University (No. QDJRCYJ003).

**Conflicts of Interest:** The authors declare no conflict of interest.

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
