# Peer review of "Models and Theoretical Analysis of SoOp Circular Polarization Bistatic Scattering for Random Rough Surface"

_remotesensing, doi:10.3390/rs12091506_

Round 1

Reviewer 1 Report

I would like to propose next corrections:

  1. Rows 78-79: The P-band signal can really penetrate to a depth of 40 cm. But this is possible at soil water content no more 2-3 volumetric %. At this moisture, there is no vegetation cover. In real conditions, at vegetation presence, the penetration depth of the P-band signal is 10-15 cm and L-band signal no more 5cm (see [7]). Correction required.
  2. Row 122: Surface-related length must be replaced by correlation length.
  3. Row 156: Mueller matrix components Svv, Svh, Shh, ..... must be named and briefly described.
  4. Row 179: The Indian regional navigation satellite system has not P band, therefore row 179 need correction.
  5. Row 184: Here, soil water content value, soil texture, and way to calculate soil dielectric constant must be added.
  6. Row 188: Here, physical explanation, obtained dependences BRCS versus frequency (fig. 2) must be added.
  7. Row 202: Figure 4 must be replaced by Figure 3.
  8. Row 202: Figure 5 must be replaced by Figure 4.

Author Response

Thank you very much. Your suggestions are very helpful for us, and we have revised our manuscript according to your suggestions. Please see the attachment.

Reviewer 2 Report

This paper presented the results of forward scattering BRCS simulation on land rough surfaces for various soil moistures, polarizations, scattering models and geometries. The results showed are meaningful for the future mission, so the manuscript is suitable for publication. The reviewer has some minor comments.

  1. Page 2, line 58: Does 0.775Ghz represent the P-band?
  2. Check the equations (8) and (10).
  3. In general, the first subscript p of the scattering amplitude S_pq refers to the polarization component of the scattered wave intercepted by the receive antenna and the second subscript q refers to the polarization of the incident wave. However, the authors used the opposite way in the manuscript. Please confirm the notations and leave a comment for the readers.
  4. From Figure 2, the authors chose 0.3 GHz frequency for the analysis. Is there any specific reason?
  5. Regarding the soil moisture properties, which models were deployed for the simulation?
  6. Regarding Figure 3, in general, the major portion of the forward scattering reflections is from the specular scattering with the in-plain geometry. The authors chose the off-specular case for this plot, but it would be good to compare the results between specular, near-specular, and off-specular cases if possible. There is also a 2-D plotting option.
  7. Page 6, line 196: The authors mentioned “the change of BRCS with soil moisture at P-band, 30 ° incident angle, 5 ° reflection zenith angle, and 120 ° scattering azimuth angle” is plotted for Figures 4 and 5. What changes of BRCS with soil moisture? Need to be explained more in detail. However, the y-axis labels for plots are just “BRCS”. If it shows BRCS, soil moisture should be fixed. Please confirm and comment.
  8. To see the sensitivities well, the same y-axis range is suggested for Figures 4 and 5. (also Figure 3 if possible.)
  9. Instead of case analysis using 1-D plots, 2-D plots, as shown in [13], could be applied in future analysis.
  10. Check the typos (i.e. Mu should be M (line 156 in Page 5)) and grammar in the context.
  11. Need to correct the references. They have formatting issues and include wired [J], [C], [M], and so on.   

Author Response

Thank you very much for your reiview. We have carefully modified our manuscript according to you valuable suggetions. Please see the attachment.

Reviewer 3 Report

Introduction

Remarks: The introduction is well described. It presents the problem well. Well-founded

Theory

Remarks: The theory is not sufficiently detailed by the authors. The theoretical basis needs great attention from the authors. It is necessary to describe the variables, the parameters, as well as more details for the equations.

Figures

Figure 3: How the authors explain water content values ​​greater than 60%.

General comments on the figures: Insert all graphs on the same scale.

Pag 5. Line 169.: ’Simulation and Analysis’ this part needs more detail.

Results
In general, the results are not sufficiently confronted with the literature. The discussion needs to be further developed. The discussion is not sufficiently supported by the literature. The authors can take advantage of the good review made in the introduction to discuss the results.
The interpretation of the data itself can be carried out in more depth.

Conclusion

In my opinion,  the conclusion presented is very general. I recommend pointing out the main differences observed between the sensors from a practical point of view.

Author Response

Thank you very much. Your suggestions and comments are very valuable for us. we have revised our manuscript according to your comments. Please see the attachment.

Round 2

Reviewer 3 Report

Dear authors, in my opinion, the article is ready for publication. Thank you very much for the replies.